# Autotaxin in Breast Cancer: Role, Epigenetic Regulation and Clinical Implications

**DOI:** 10.3390/cancers14215437

**Published:** 2022-11-04

**Authors:** Andrianna Drosouni, Maria Panagopoulou, Vassilis Aidinis, Ekaterini Chatzaki

**Affiliations:** 1Laboratory of Pharmacology, Medical School, Democritus University of Thrace, 68100 Alexandroupolis, Greece; 2Institute of Agri-Food and Life Sciences, Hellenic Mediterranean University Research Centre, 71410 Heraklion, Greece; 3Institute of BioInnovation, Biomedical Sciences Research Center Alexander Fleming, 16672 Athens, Greece

**Keywords:** autotaxin, *ENPP2*, lysophosphatidic acid, ATX-LPA signaling, expression, methylation, breast cancer, epigenetics

## Abstract

**Simple Summary:**

Autotaxin (ATX) has been linked with the pathogenesis of several cancers and especially with breast cancer (BC). BC is one of the most common cancers among women and although significant steps have been made regarding its early detection and treatment options, challenges still remain. This review aims to summarize the current knowledge of the role and regulation of ATX in BC and to shed light on its potential for clinical applications.

**Abstract:**

Autotaxin (ATX), the protein product of Ectonucleotide Pyrophosphatase Phosphodiesterase 2 (*ENPP2*), is a secreted lysophospholipase D (lysoPLD) responsible for the extracellular production of lysophosphatidic acid (LPA). ATX-LPA pathway signaling participates in several normal biological functions, but it has also been connected to cancer progression, metastasis and inflammatory processes. Significant research has established a role in breast cancer and it has been suggested as a therapeutic target and/or a clinically relevant biomarker. Recently, *ENPP2* methylation was described, revealing a potential for clinical exploitation in liquid biopsy. The current review aims to gather the latest findings about aberrant signaling through ATX-LPA in breast cancer and discusses the role of *ENPP2* expression and epigenetic modification, giving insights with translational value.

## 1. Introduction

Autotaxin (ATX) is a secreted glycoprotein that was first isolated by Stracke’s lab in 1992 in human melanoma cells [1]. ATX belongs to the Ectonucleotide Pyrophosphatase/Phosphodiesterase (*ENPP*) family, encoded by *ENPP2* [2]. Expression of ATX is reported in many tissues and biological fluids and it is considered to be responsible for the production of the circulating lysophosphatidic acid (LPA) [3]. It is now well-established that the biological effects of ATX emerge from the production of LPA and ATX-LPA axis signaling [4]. Notably, it was observed that *ENPP2*^+/−^ mice had reduced LPA in plasma by half in comparison to normal controls [5]. Although ATX is found in human circulation, its action is more local than systemic, due to its short half-life in blood [6].

In health, ATX is related with embryonic development and wound healing. However, growing evidence has linked the ATX-LPA axis to several diseases but also to cancer [6,7,8]. Aberrant ATX-LPA signaling seems to affect tumor progression, metastatic potential and invasiveness [9]. In fact, *ENPP2* is considered one of the top 40 genes that promote the metastatic process [10]. Importantly, ATX and LPA have been recognized as potential diagnostic biomarkers and drug targets for cancer and chronic inflammatory diseases [11]. Recently, our studies have shown that at the epigenetic level, methylation could play a key role in ATX regulation and pathogenesis [12,13]. Below, we present a review of the literature regarding the role of ATX and *ENNP2* expression in the malignancy that it is more studied, i.e., breast cancer (BC), focusing on its potential implementation as a liquid biopsy biomarker. Finally, we highlight its potential for clinical application as a therapeutic target and prognostic biomarker in other cancer types.

## 2. ATX Structure and Function

At the gene level, *ENPP2* is located on chromosomal region 8q24 and contains 26 introns and 27 exons [14]. It is characterized by alternative splicing of mRNA and so far, five splice variants have been identified, namely α, β, γ and most recently δ and ε [15]. All of the distinct variants are catalytically active with lysoPLD activity, but their isoform specific functions are not yet characterized [16,17]. Each isoform presents differences in stability, while their expression pattern also differs among tissues [14].

At the protein level, ATX is a type II ectonucleotide of 125 kDa [18]. When ATX was first discovered, it was characterized as a cell motility factor, whilst now, it is included in the *ENPP* family, which consists of seven structurally related ectoenzymes [2,19]. ATX functions as a lysophospholipase D (lysoPLD) that mainly hydrolyzes extracellular lysophosphatidylcholine (LPC) to generate LPA plus choline and is also the only LysoPLD in the *ENPP* family [2]. LPA is a bioactive phospholipid, which is widely expressed in many different tissues and acts via six G-protein coupled receptors (LPAR1-6) which can activate several different signaling pathways in physiological and pathological conditions [20]. Besides its ability to hydrolyze lysophospholipids, such as LPC into LPA, ATX can also hydrolyze phosphosphingolipids, such as sphingosylphosphorylcholine (SPC) into sphingosine 1-phosphate (S1P), and nucleotides, still showing higher affinity to LPC; it is clear that by generating different products, ATX can lead to different signaling results [20,21,22,23]. In addition, it is important to mention that the actions of ATX are restricted by liver sinusoidal endothelial cells, which rapidly clear ATX from the circulation [24].

Structurally, ATX is a multi-domain protein, first synthesized as a pre-proenzyme that undergoes maturation for a full active enzyme to be produced [25,26]. ATX consists of a central catalytic phosphodiesterase domain (PDE) which interacts on one side with two N-terminal somatomedin-like domain regions (SMB1-2) and on the other side with the catalytically inactive N-terminal nuclease type (NYC) domain [27]. Importantly, the formation of a deep hydrophobic pocket in the catalytic domain of ATX which bounds the substrate LPC, not found in any other phospholipase, seems to be responsible for ATX’s activity as a lysoPD [26,28].

Notably, the structure of ATX allows the ability to bind to the cell surface and directly deliver LPA to cells. In particular, ATX loaded by LPA, can bind to the surface of exosomes and be transported to the target cells, there it can interact with adhesive molecules such as integrins and bind to the cell surface releasing LPA and promoting cell specific signaling [27,29,30]. By binding to cells, ATX can regulate LPA production, enhance signaling responses specific to cells and also protect LPA from degradation [10,31].

## 3. ATX-LPA Signaling 

As already mentioned, most of the circulating LPA is generated by ATX [32,33]. LPA is a bioactive lipid that is produced both extracellularly and intracellularly and has been correlated with a variety of biological functions [34]. Normally, it can be detected in small amounts in all eukaryotic tissues with the highest levels in the brain and it is also detected in most biological fluids, with high concentration in blood plasma, while being one of the major active constituents in serum [11,20].

LPA signals through its LPAR1-6 receptors, being widely expressed in many different tissues and organs, where they regulate a broad range of cellular processes in both physiological and pathological conditions [20]. The *LPAR1,2* and *3* genes are included to the subfamily of Endothelial Differentiation Gene (*EdG*), while *LPAR4 and LPAR5* are included to the purinergic subfamily [35]. More recently, the novel GPC receptor P2Y5 was detected, which was named *LPAR6* [36]. It has been found that LPA can also bind to non-GPC receptors, although most LPA functions are known from binding with the GPCRs [8,37]. The first in vivo biological role that was attributed to signaling through LPA was the regulation of blood pressure [38]. Since then, many biological responses have been attributed to the ATX-LPA axis such as angiogenesis, re-epithelialization, cell proliferation and migration, cell differentiation, platelet activation and aggregation, neurite remodeling and ion channel activation, enhanced cell survival and induced inflammation by the production of pro-inflammatory cytokines and also mitogenic and chemotactic activities [17,20,26,34]. Those diverse and widespread results of the ATX-LPA signaling are greatly influenced by the heterogeneity of the LPA receptors, the different expression patterns, tissue context, selective activation and also the different pathways involved in signaling [11]. Meanwhile, concentration levels of LPA are regulated primarily by ATX that controls its production rate and then by LPPs which are responsible for its degradation [39].

## 4. Regulation of ATX Production and Activity

*ENPP2* is expressed in many tissues, with high levels in brain and adipose tissue, and is also found in many biological fluids with high concentration in blood and serum [20]. Regulation of ATX production is crucial, as it is linked with several cancer types and inflammatory conditions [10,40], but it is still not yet fully understood how the ATX production is regulated due to the complicated mechanisms and the different stages that are involved in the regulation of ATX (Figure 1).

At the transcriptional level, it has been reported that *ENPP2* expression is regulated by several Transcription Factor (TFs), some participating in carcinogenic cascades. To begin with, *ENPP2* has been reported to be activated by the *STAT3* leading to increased migration of BC cells [41]. On the other hand, reduced *ENPP2* expression is observed after decreased expression of the *NFAT1*, resulting in a reduction of growth and metastasis of melanoma cells [42]. AP-1 and SP seem to act on the *ENPP2* promoter affecting its transcription in human neuroblastoma cell lines [43], whereas v-jun and c-jun also seem to hold a role [44,45]. In addition to TFs, a regulating role at transcriptional level has been attributed to the Hypoxia Inducible Factors (HIFs). It has been shown that in Hepatocellular carcinoma (HCC), HIFs increase *ENPP2* mRNA and ATX protein expression, which further supports HCC progression [46]. Signaling pathways have also been linked with *ENPP2* expression. In fact, RSPO2, a regulator of the Wnt/b-catenin signaling, has been reported to act directly on *ENPP2* and increase its expression in myoblast cells [47]. *ENPP2* expression was also induced by type I interferons through activation of the signaling pathways JAK-STAT and PI3K-AKT [48]. Furthermore, after inhibition of the AKT pathway, it was found that it also has a role in secretion of ATX [49].

At the post-transcriptional level, ATX is regulated by RNA-binding proteins HuR and AUF1. HuR acts through stabilizing the *ENPP2* mRNA and leads to increased *ENPP2* expression, while AUF1 promotes *ENPP2* decay and suppress *ENPP2* expression; it was suggested that via these mechanisms they can modulate cancer cell migration [50]. In addition, it has been shown that the tumor suppressor microRNA-101-3p could bind to the 3′-UTR of *ENPP2* mRNA and inhibit its translation, leading to the reduction of migration, invasion and proliferation [51]. Another study showed that the 3′-UTR of *ENPP2* mRNA could also be targeted by the RNA methyltransferase NSUN2 and its methylation seems to promote the mRNA export from the nucleus leading to enhanced translation of ATX [52].

The secretion of ATX is also a regulated process. ATX is synthesized as a pre-pro-enzyme and in order to produce a mature full activated enzyme it undergoes firstly cleavage of a N-terminal signal peptide between amino-acids G27 and G28, and then N-glycosylation at the amino acids N53 and N410, those steps being both required for the secretion of ATX. In addition to that, N-glycosylation of the site Asn-524 has been implicated in ATX’s activity as a lysoPLD and as a chemoattractant [53,54,55]. After secretion, the activity of ATX seems to also be regulated through a negative feedback loop that is created because the ATX products, LPA and S1P, can bind to ATX more strongly than LPC [40,56]. However, it has been suggested that inflammatory cytokines are able to counteract the inhibition of *ENPP2* mRNA expression by S1P and LPA [40].

## 5. The Role of Methylation in the Regulation of *ENPP2* Expression

Recently the epigenetic changes of *ENPP2* have been investigated and a few studies showed that *ENPP2* expression can be regulated by DNA methylation (Table 1). DNA methylation is probably the most studied epigenetic alteration in mammals, as it provides a vital mechanism for stable gene silencing [57,58]. DNA methylation occurs with the addition of a methyl group solely in the 5 position of the cytosine ring and is almost inclusively seen in CpG dinucleotides (CGs) which are called “CpG islands” [57,59].

In our recent analysis, we studied the methylation of *ENPP2* in multiple healthy tissues via a bioinformatic in silico approach and correlated it with gene and isoform expression. We examined publicly available high-throughput methylation datasets from studies involving the Illumina methylation bead-chip arrays found in Gene Expression Omnibus (GEO), to identify Differentially Methylated CpGs (DMCs) of *ENPP2* [12]. We showed a consistent methylation pattern throughout the *ENPP2* gene across 17 healthy human tissues. In particular, increased methylation was observed in the gene body CGs and decreased methylation in the promoter and the 1st exon CGs. Given the fact that *ENPP2* is abundantly expressed in many tissues and biological fluids [66], we suggest that the decreased methylation in the promoter is associated with the active transcription of the gene in healthy tissues.

## 6. Role of ATX in Breast Cancer

Since the discovery of ATX as a motility factor, many studies have focused on its role in cancer [1]. ATX expression and subsequent LPA production are elevated or aberrant in both tumor and serum from almost every type of cancer, such as breast [67], thyroid [68], HCC [69], lung [70], renal [71], pancreatic [72], bladder [73], ovarian [3], endometrial [74], prostate [75], glioblastoma [76] and neuroblastoma [77]. Consequently, ATX and LPA have been proposed as diagnostic biomarkers and drug targets, with some inhibitors already in phase I and II of clinical trials [11,23,72].

ATX is strongly related with enhanced proliferation, migration, and survival of cancer cells [9]. The first link of ATX with tumorigenesis came from a study on Ras transformed NIH3T3 cells that overexpressed ATX and enhanced tumor growth, angiogenesis and aggression of cancer cells, while inactive mutant ATX did not [78,79]. Consistent with this, enhanced expression of ATX and production of LPA by Epstein–Barr virus infection promoted growth and survival in Hodgkin lymphoma cells. In addition, specific downregulation of ATX led to decreased LPA levels resulting in reduced cell growth and viability in these cells [80]. Interestingly, transgenic mouse models that expressed either human ATX, LPAR1, LPAR2 or LPAR3 displayed spontaneous development of breast tumors [9], while in human BC MDA-B0-2 cells, overexpression of ATX or LPAR1 resulted in invasion, bone metastasis and destruction [81,82]. Generally, ATX-LPA axis signaling is proposed to highly affect BC-related inflammation and consequently progression [83]. Notably, inhibition of *ENPP2* led in the decrease of lung and bone metastasis but did not seem to affect the progression of the breast primary tumor [82]. Inhibition of LPAR1 also had similar results [84].

However, ATX mRNA levels in BC tumor biopsies are not good indicators for cancer metastasis and progression [82]. Intriguingly, ATX protein was found to be stored in a-granules of resting human platelets [85]. BC cells co-cultured with platelets showed an increase in LPA and cancer cell proliferation, indicating that interaction between cancer cells and platelets promotes platelet activation. Furthermore, inhibition of integrin αIIbβ3 in cancer cells prevented platelet degranulation and release of ATX leading to reduced cell proliferation [85]. These findings imply that non-tumor ATX in circulation might derive from platelets after interaction with circulating tumor cells, affecting metastasis, a pathway that could present a therapeutic target.

In addition, as mentioned previously, cell-secreted exosomes can bind to ATX, which provides a mechanism for extracellular LPA production and LPA delivery to cell surface receptors, protecting LPA from degradation. Importantly, this mechanism also seems to stimulate LPA signaling in cells and promote exosome-stimulated cell migration [29].

*ENPP2* expression can be altered by several inflammatory cytokines and growth factors, frequently increased in cancer [86]. TNFα/NF-kB has been associated with increased *ENPP2* expression and activity in human Hep3B and Huh7 liver cancer cells [69]. On the contrary, decreased ATX activity was caused by the cytokines IL-1β, TGF-β and IL-4 in thyroid carcinoma UTC-1736 cells, while IL-6 enhanced *ENPP2* expression [68]. In addition, a concentration-dependent effect was observed from the growth factors EGF and bEGF that caused increased mRNA expression of *ENPP2* [68]. Finally, it was proposed that induction of *ENPP2* expression by VEGF was associated with an aggressive phenotype in ovarian cancer via a positive feedback loop [87]. All these factors are dysregulated in BC and could form pathophysiological loops with ATX, needing further assessment.

ATX-LPA axis signaling was also proposed to cause resistance in chemotherapy and radiotherapy, by protecting cancer cells against cell death caused by therapy [56]. Similar to findings in ovarian cancer cells lines, where expression of ATX was linked to a delay in chemotherapy-induced apoptosis [88], increased levels of *ENPP2* mRNA and inflammatory mediators were observed in adipose tissue during γ-radiation, suggesting activation of the inflammatory response cycle, which protected BC cells and reduced the efficiency of radiotherapy [89]. Similarly, inhibition of ATX blocked the protection from apoptosis induced by the chemoattractant Taxol in the BC cell line MCF-7 [90]. Notably, it was also observed that radiation could induce the production of ATX, associating ATX with DNA damage in BC, while dexamethasone can attenuate the radiation-induced increase of ATX [91]. Drug resistance was also observed in Breast Cancer Stem like Cells (CSCs) and was attributed to ATX. In breast CSCs, ATX inhibitors 30 and 3b were combined with the chemotherapeutic drug Paclitaxel (PT), resulting in 25–30% reduced cell viability as compared to PT alone, suggesting that ATX is implicated in drug resistance and its inhibition can resensitize CSCs [92]. Similar observations were also found in ovarian cancer [93]. Additionally, ENPP2 was identified as the second most upregulated gene in breast CSCs after treatment with PT in an in silico analysis, indicating that CSCs could favor LPA-enhanced microenvironment [94].

All these results give strong evidence that pathological signaling through the ATX-LPA axis and aberrant expression of *ENPP2* is linked to poor outcome in BC. However, as a word of caution, although increased ATX levels is correlated with poor outcome, it does not automatically reflect LPA levels and signaling, as many factors such as LPC availability and LPA degradation can affect LPA levels [4]. Eventually, the LPA expression and signaling on both the tumor and surrounding stromal cells will determine the results of ATX expression [6]. In fact, it is now established that the direct production of *ENPP2* mRNA by cancer cells is significantly low in many cancer types and the majority of ATX is produced from the tumor microenvironment [95,96]. Recent studies in both thyroid and BC revealed a prototypical model for the crosstalk between the tumor microenvironment and ATX expression. Particularly, inflammatory mediators from cancer cells increased LPA and ATX levels in surrounding fibroblasts and adipose tissue that further induce the inflammation and progression of the tumor, creating a feedback cycle [97,98]. In a recent study in BC cell line models, *ENPP2* mRNA expression levels decreased in the more aggressive MDA-MB-231 line as compared to the much less aggressive MCF-7 [60]. In addition to that, in human lung-cancer tissue samples, *ENPP2* mRNA was founbred significantly down-regulated, in both in silico and experimental analysis. However, the ATX protein expression in tissues and activity in serum were increased compared to control samples. Identical findings were found in two lung cancer mouse models. Those results were proposed to arise from integrin binding of ATX to cancer cells that protected it from degradation [99]. Taken together, these results suggest that several mechanisms are implicated in mRNA/protein *ENPP2* expression and LPA signaling in tumors during cancer pathogenesis, building a complex and possibly tissue-specific process.

## 7. *ENPP2* Methylation and Cancer

Abnormal methylation detected in tissue or liquid biopsy has been associated with cancer development and progression [100,101,102,103,104,105]. Several studies address the regulation of ATX expression at its gene methylation level in cancer and may enlighten different aspects of its involvement in the pathogenetic process in BC (Table 1). A relevant work showed that *ENNP2* was highly methylated in BC tissues, also presenting low mRNA expression levels compared to adjacent tissues [60]. Furthermore, in silico studies based on TCGA datasets reported high methylation of *ENPP2* in BC and characterized it as a methylation-driven gene in cancer [61,62]. In our recent in silico analysis of *ENPP2* methylation in BC tissues, it was found that the promoter CGs were hypermethylated in comparison to normal tissues and mRNA expression was downregulated and inversely correlated with the methylation. Hypermethylation of *ENPP2* was also associated with the progression of BC [13]. Together, these findings suggest that *ENPP2* is highly methylated in BC, correlated with aggressiveness and low expression levels. Interestingly, Wang et al. demonstrated that the promoter region of *ENPP2* was hypermethylated in tissue samples from BC. Furthermore, in an analysis of circulating cell free DNA (ccfDNA), *ENPP2* was also found hypermethylated but failed to find any significant difference between methylation levels of healthy and BC patients [63]. In contrast to that, in our recent work, methylation of *ENPP2* 1st exon in ccfDNA was found significant hypermethylated in BC patients as compared to healthy ccfDNA and it was also correlated with the cancer load [13], suggesting a role of *ENPP2* methylation as a BC prognostic biomarker. These controversial results could be methodologically explained due to the different regions of *ENPP2* gene that were examined in the patient samples. In addition, the difference in sample size might explain the failing to demonstrate statistical significance in differential methylation. In silico analysis of *ENPP2* conducted in different cancer types such as Pancreatic cancer (PC), LC, Colorectal cancer (CC), melanoma and HCC revealed a specific methylation pattern, as *ENPP2* was hypermethylated in promoter and hypomethylated in gene body, accordingly. Most importantly, decreased expression levels were associated with increased promoter methylation. Additionally, in the same study, methylation was related to poor prognostic parameters [12]. In another in silico analysis, *ENPP2* was found hypermethylated in Lung Adenocarcinoma (LUAD) and Squamous Cell Carcinoma (SCC) tumors and this was consistent with lower mRNA expression, highly associated with worse progression in LUAD [64]. Consistent with that, a study investigating epigenetic changes reported *ENPP2* promoter hypermethylation in metastatic Uveal Melanoma (UM) that was associated with downregulation in mRNA expression [65]. The above studies demonstrate that hypermethylation of *ENPP2* promoter and 1st Exon in cancer is correlated with reduced ATX expression, presenting an epigenetic expression regulation level, and suggests a potential of *ENPP2* methylation as a prognostic biomarker, awaiting further validation.

As mentioned earlier, although elevated *ENPP2* levels have been observed in cancer patients, most of the produced *ENPP2* is proposed to originate from the tumor surrounding environment rather than directly from the cancer cells themselves [95,96]. This could provide an explanation for the increased methylation and consecutively decreased expression that is mainly observed in cancer cells.

## 8. *ENPP2* Methylation in BC Liquid Biopsy

Assessing methylation in liquid biopsy biomaterial such as ccfDNA can dynamically reflect methylation events of the tumor, as supported by in vitro studies showing that the methylation profile of ccfDNA released by BC cell lines in culture is identical to their genomic DNA [106]. ccfDNA has, therefore, emerged as a valuable source of clinically relevant information and it is currently exploited in multiple applications for detecting epigenetic biomarkers for prognosis, diagnosis and disease monitoring in BC [102,107]. In parallel to studies in BC tissues, *ENPP2* hypermethylation of promoter associated CGs was also demonstrated in silico. Thus, datasets of ccfDNAs from BC patients showed higher methylation in relation to ccfDNAs from healthy individuals [13]. We further evaluated *ENPP2* methylation in ccfDNA of BC patients in order to examine its clinical value as a biomarker. Analysis of the 1st Exon cg02534163 by a targeted qMSP assay showed that *ENPP2* hypermethylation was detected more often in ccfDNA of BrCa patients than in healthy individuals [13]. However, in a previous study addressing *ENPP2* methylation in ccfDNA from 22 healthy and 45 Taiwanese BC patients, no significant differences were reported [63], although *ENPP2* methylation showed a two-fold increase in BC in relation to adjacent normal tissue. Methodological differences or even population genetic variations might explain these different findings. Importantly, in our study, ccfDNA methylation levels of *ENPP2* were also elevated in the neoadjuvant and metastatic groups of patients in relation to adjuvant and control group of patients. This result could be due to the fact that in the neoadjuvant and metastatic groups, patients still have a significant tumor burden. Our bioinformatic analysis also showed that *ENPP2* methylation was increased in metastasis in relation to primary cancers. In addition to that, according to our experimental analysis, patients having two or more metastatic foci presented more increased *ENPP2* methylation levels than those patients having a distant metastasis in one focus [13]. Cumulative experimental results are in accordance with those from bioinformatic analysis showing hypermethylation of *ENPP2* in both BC tissue and ccfDNA and a correlation with cancer aggressiveness and metastasis, suggesting its potential as a novel circulating biomarker in BC.

Circulating microRNAs are also promising biomarkers, as they often present a distinct expression profile in cancers [108]. An interesting case is miR489-3p, as it was found significantly increased in the serum of mouse models with ATX-induced tumors, and in an in silico analysis of human cancers’ serum [109]. In general, miR-489-3p has a tumor suppressing role and acts by suppressing mitogen-activated protein kinase (MEK1), a protein implicated in tumor development and progression [110]. However, it was observed that expression of ATX could alter the tumor suppressive function of miR-489-3p in tumor cells and enhance MEK1 activity and consecutively tumor appearance [109].

## 9. The ATX-LPA Axis as a Therapeutic Target in BC

Due to the established implication of ATX in cancer, blocking the ATX-LPA axis signaling could present an important target of therapeutic intervention. Most importantly, ATX is considered a strongly druggable target as the action of its PDE domain is easily inhibited, and ATX also acts extracellularly; therefore, it has gained a lot of attention from both academic and industrial settings with several ATX inhibitors being in clinical or pre-clinical studies for drug development [26]. In principle, inhibitors of ATX function could be directed towards its synthesis, maturation and/or its catalytic activity [25]. Initially, many lipid analogs were designed based on the fact that LPA can inhibit the LysoPLD activity of ATX [110]. One of the most successful was the inhibitor BMP-22, an ATX inhibitor, which resulted in reduced lung metastasis in melanoma; while it was also shown that it can reduce bone metastasis in BC [85,111]. However, the bioavailability and efficiency of lipid analogs is limited in vivo [110]. After the discovery of ATX structure, potent non-lipid inhibitors have been designed with great success. Indeed, the competitive inhibitor of ATX, GLPG1690, was originally designed for the IPF treatment reaching phase III clinical trial [112]. Unfortunately, the clinical trials (NCT03711162, NCT03733444), were recently discontinued due to the benefit-risk profile. GLPG1690 was also studied in BC resulting in decreased proliferation of cancer cells, while it promoted apoptosis induced by radiotherapy when combined together [113,114]. Another promising ATX inhibitor is ONO-8430506, which inhibited breast tumor growth and subsequent blocked metastasis in liver and lung. ONO-8430506 has also resulted in reduced tumor growth in thyroid cancer [97,115]. In addition, the LPA receptor antagonist BrP-LPA was studied for the treatment of BC, leading to reduced cancer cell migration and tumor regression [116]. Other ATX inhibitors such as IOA-289 and PF8380, or BMS-986020, a LPAR1 antagonist, were developed and showed promising results in preclinical and clinical trials in other conditions [117,118,119,120]; however, they have not yet been tested in BC. Given the importance of ATX-LPA signaling, expanding research of these novel molecules in BC could lead to new therapeutic alternatives.

## 10. Conclusions

*ENPP2* is responsible for the production of circulating LPA, and it is well established that the actions of ATX emerge from ATX-LPA axis signaling through LPAR1-6. Aberrant levels of ATX have been found in BC, where its production seems downregulated and hypermethylated in tumor cells and overexpressed in surrounding tissue. It is quite clear by recent findings that methylation of *ENPP2* at the promoter region or 1st exon holds a significant role in the regulation of its expression, and it has been proposed that *ENPP2* methylation can serve as a biomarker for BC diagnosis and prognosis with the potential to be implemented in liquid biopsy, i.e., detected in ccfDNA, baring additional clinical value. In addition, ATX activity has a significant role in BC progression, invasion and metastasis, opening the way for new treatment strategies based on *ENPP2* inhibition.

## Figures and Tables

**Figure 1 cancers-14-05437-f001:**
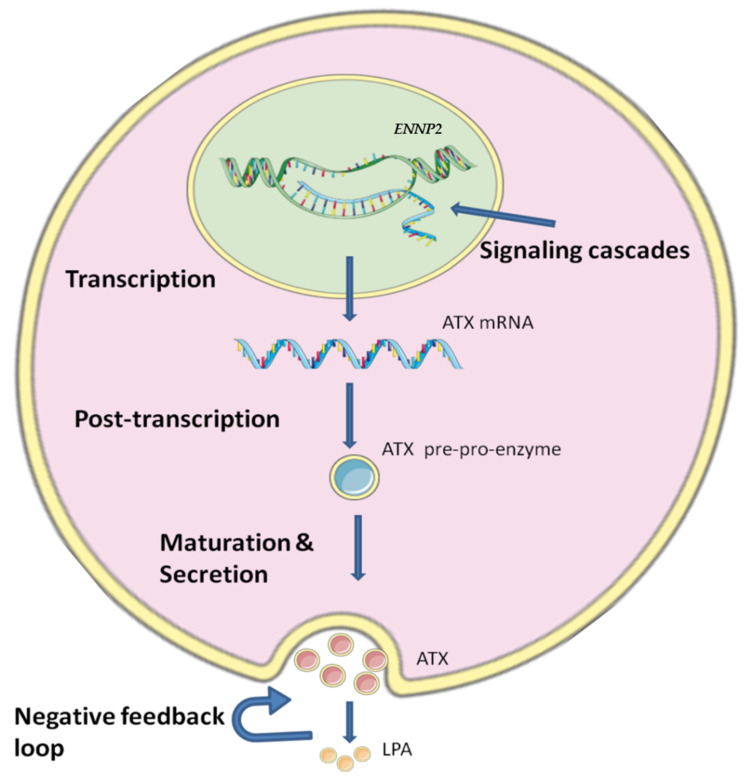
The regulatory levels of ATX’s expression and production. The expression of ATX is regulated in many ways. The first step of regulation for *ENPP2* expression is during transcription by methylation and different transcription factors, growth factors and cytokines which can modulate *ENNP2* expression by a diversity of signaling cascades. Then, at post-transcriptional level, a variety of factors act on *ENNP2* mRNA affecting its stability and translation. Maturation and secretion also have an important role in regulating ATX, as a negative feedback loop is formed by ATX’s product LPA. The figure was prepared using SMART (smart.servier.com, accessed on 5 July 2022).

**Table 1 cancers-14-05437-t001:** Methylation status of *ENPP2* in cancer in relation to health.

Disease	Group Size	Biological Material	Gene Area	Methylation Status	Expression Levels	Reference
BC	783 tumor vs. 109 paracancerous	Tissue	Promoter	Hypermethylated	Decreased	[60]
BC	557 cancer vs. 90 normal	Tissues	MicroRNA target site	Hypermethylated	N/D	[61]
BC	796 tumor vs. 96 normal	Tissue	N/D	Hypermethylated	N/D	[62]
BC	−109 tumor vs 109 normal tissues, 45 cancer vs. 22 normal ccfDNAs	Tissue & ccfDNA	Promoter	Hypermethylated	N/D	[63]
BC	520 tumor vs 185 normal tissues, 86 cancer vs. 46 normal ccfDNAs	Tissue & ccfDNA	1st exon	Hypermethylated	Decreased	[13]
LC	473 LUAD vs 320 SCC vs 52 normal	Tissue	Promoter	Hypermethylated	Decreased	[64]
PC, LC, CC, melanoma and HCC	73 PC vs. 63 prostate benign, 17 LC vs. 43 adjacent, 252 CC vs. 252 adjacent, 89 melanoma vs. 73 nevus, 30 HCC vs. 30 adjacent, 19 primary HCC vs. 18 recurrent, HCC vs. 18 adjacent, 212 LC vs. 15 healthy, 235 PC vs. 35 healthy, 241 HCC vs. 42 healthy	Tissue	Promoter	Hypermethylated	Decreased	[12]
Uveal melanoma	10 metastatic vs 29 primary	Tissue	Promoter	Hypermethylated	Decreased	[65]

Abbreviations: BC: Breast Cancer, LC: Lung Cancer, PC: Pancreatic Cancer, CC: Colon Cancer, HCC: Hepatocellular Cancer, LUAD: Lung Adenocarcinoma, SCC: Squamous cell carcinoma, ccfDNA: circulating cell free DNA, N/D: not determined.

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
