# Peer review of "Autotaxin in Breast Cancer: Role, Epigenetic Regulation and Clinical Implications"

_cancers, 2022, doi:10.3390/cancers14215437_

Round 1

Reviewer 1 Report

The most prevalent form of cancer among women is breast cancer. Due to population expansion and changes in the prevalence of cancer risk factors, the burden of breast cancer incidence has risen over time. Breast cancer in particular, and other cancers with similar pathophysiology, have been related to autotaxin (ATX). Although great progress has been achieved in the areas of early identification and treatment options for breast cancer, nevertheless, challenges still exist. The authors purpose of this study is to provide an overview of the current understanding and regulation of ATX in breast cancer. The article is well written; however, I have a few concerns.

I suggest that the author write briefly about the function of exosomes binding to ATX in breast cancer physiology.

The authors should also mention the effect of ATX on breast cancer stem cells and how it regulates drug resistance mechanisms.

Author Response

We thank the reviewer for the meticulous reading of the manuscript. Now according to reviewer’ instructions, we added important information about drug resistance in lines 242-249“Drug resistance was also observed in Breast Cancer Stem like Cells (CSCs) and attributed to ATX. In breast CSCs ATX inhibitors 30 and 3b were combined with the chemotherapeutic drug Paclitaxel (PT), resulting in 25-30% reduced cell viability as compared to PT alone, suggesting that ATX is implicated in drug resistance and its inhibition can resensitize CSCs[1]. Similar observations were also found in ovarian cancer[2]. Additionally, ENPP2 was identified as the second most upregulated gene in breast CSCs after treatment with PT in an in-silico analysis, indicating that CSCs could favor LPA-enhanced microenvironment[3].”

Also, in lines 81-86: “Notably, the structure of ATX allows the ability to bind to the cell surface and directly deliver LPA to cells. In particular, ATX loaded by LPA, can bind to the surface of exosomes and be transported to the target cells, there it can interact with adhesive molecules like integrins and bind to the cell surface releasing LPA and promoting cell specific signaling [28,30,31]. By binding to cells, ATX can regulate LPA production, enhance signaling responses specific to cells and also protect LPA from degradation [10,32]” and in lines 218-221 “In addition, as mentioned previously cell-secreted exosomes can bind to ATX, which provides a mechanism for extracellular LPA production and LPA delivery to cell surface receptors, protecting LPA from degradation. Importantly, this mechanism also seems to stimulate LPA signaling in cells and promote exosome-stimulated cell migration (30).

Reviewer 2 Report

In this manuscript, the authors have discussed the role of Autotaxin (ATX) in breast cancer. They described the location, structure, and functions of ATX, then the regulation of its production and activity. They further elucidate the role of ATX in breast cancer and current therapeutic drugs targeting ATX-LPA axis. Overall, they fully summarized the research of ATX in breast cancer.

Author Response

Thank you very much for your nice comments on our article